

# First report of fatty acids in *Mimosa diplotricha* bee pollen with *in vitro* lipase inhibitory activity

Phanthiwa Khongkarat[1], Prapun Traiyasut[2], Preecha Phuwapraisirisan[3] and Chanpen Chanchao[4]

[1] Program in Biotechnology, Faculty of Science, Chulalongkorn University, Bangkok, Thailand
[2] Program in Biology, Faculty of Science, Ubon Ratchathani Rajabhat University, Ubon Ratchathani, Thailand
[3] Department of Chemistry, Faculty of Science, Chulalongkorn University, Bangkok, Thailand
[4] Department of Biology, Faculty of Science, Chulalongkorn University, Bangkok, Thailand

Corresponding author
Chanpen Chanchao,
chanpen.c@chula.ac.th

## ABSTRACT

Bee pollen (BP) is full of nutrients and phytochemicals, and so it is widely used as a health food and alternative medicine. Its composition and bioactivity mainly depend on the floral pollens. In this work, BP collected by *Apis mellifera* with different monoculture flowering crops (BP1-6) were used. The types of floral pollen in each BP were initially identified by morphology, and subsequently confirmed using molecular phylogenetic analysis. Data from both approaches were consistent and revealed each BP to be monofloral and derived from the flowers of *Camellia sinensis* L., *Helianthus annuus* L., *Mimosa diplotricha*, *Nelumbo nucifera*, *Xyris complanata*, and *Ageratum conyzoides* for BP1 to BP6, respectively. The crude extracts of all six BPs were prepared by sequential partition with methanol, dichloromethane (DCM), and hexane. The crude extracts were then tested for the *in vitro* (i) $\alpha$-amylase inhibitory, (ii) acetylcholinesterase inhibitory (AChEI), and (iii) porcine pancreatic lipase inhibitory (PPLI) activities in terms of the percentage enzyme inhibition and half maximum inhibitory concentration ($IC_{50}$). The DCM partitioned extract of *X. complanata* BP (DCMXBP) had the highest active $\alpha$-amylase inhibitory activity with an $IC_{50}$ value of 1,792.48 $\pm$ 50.56 µg/mL. The DCM partitioned extracts of *C. sinensis* L. BP (DCMCBP) and *M. diplotricha* BP (DCMMBP) had the highest PPLI activities with an $IC_{50}$ value of 458.5 $\pm$ 13.4 and 500.8 $\pm$ 24.8 µg/mL, respectively), while no crude extract showed any marked AChEI activity. Here, the *in vitro* PPLI activity was focused on. Unlike *C. sinensis* L. BP, there has been no previous report of *M. diplotricha* BP having PPLI activity. Hence, DCMMBP was further fractionated by silica gel 60 column chromatography, pooling fractions with the same thin layer chromatography profile. The pooled fraction of DCMMBP2-1 was found to be the most active ($IC_{50}$ of 52.6 $\pm$ 3.5 µg/mL), while nuclear magnetic resonance analysis revealed the presence of unsaturated free fatty acids. Gas chromatography with flame-ionization detection analysis revealed the major fatty acids included one saturated acid (palmitic acid) and two polyunsaturated acids (linoleic and linolenic acids). In contrast, the pooled fraction of DCMMBP2-2 was inactive but pure, and was identified as naringenin, which has previously been reported to be present in *M. pigra* L. Thus, it can be concluded that naringenin was compound marker for *Mimosa* BP. The fatty acids in BP are nutritional and pose potent PPLI activity.

## INTRODUCTION

Bee pollen (BP) is one of the economic bee products. It is derived from the flower's male gametophyte produced within anther sacs in the flowers of angiosperms, and is collected by foragers of bees and mixed with the plant's nectar, wax, and bee's saliva to compact the powder into pollen grains. The BP is then loaded into a pollen basket (corbicula), which is part of the tibia on the hind legs (*Costa et al., 2019*), and later stored in the hive as an essential food for the larva and adults in addition to honey, bee bread, and royal jelly. Nutritionally, BP is known as a functional food for humans and due to its high content of protein and carbohydrates, along with crude fiber, amino acids, vitamins, minerals, and fatty acids (*Yang et al., 2013*). However, floral identification of BP should be done prior to the consumption as a few people display allergic and anaphylactic reactions after consumption of certain floral pollens (*Shahali, 2015*; *Jagdis & Sussman, 2012*). In particular, bee pollen composed of ash (Oleaceae), oak (Fagaceae), willow and poplar (Salicaceae), or corn (*Zea mays*) as the dominant floral pollen requires caution (*Vieths, Scheurer & Ballmer-Weber, 2002*).

In addition, BP is also composed of secondary metabolites (polyphenols and flavonoids) of plants (*Rzepecka-Stojko et al., 2015*). In folk medicine, BP has long been used as a tonic and a multipurpose remedy, and it is widely known that various phytochemicals in BP are bioactive. Chilean BP has been reported to have an antioxidative activity in both the 2,2′-azino-bis(3-ethylbenzothiazoline-6-sulfonate (ABTS) radical and 2,2-diphenyl-1-picryl-hydrazyl-hydrate (DPPH) scavenging capacity assays. Both activities were related to their polyphenols content, especially caffeic acid, coumaric acid, luteolin, and pinocembrin (*Munoz et al., 2020*).

In addition, BP harvested by the stingless bee, *Scaptotrigona affinis postica*, in Brazil showed an anti-inflammatory activity by inhibiting cyclooxygenase (COX-1 and COX-2) and reducing the edema in male *Mus musculus* mice (Swiss strain) when using the carrageenan- and dextran-induced paw edema tests (*Lopes et al., 2020*).

However, the composition and phytochemicals in BP mainly depend on the bee species, and its botanical and geographical origins. For example, alkaloids in the BP of *Catharanthus roseus*, saponins in the BP of *Momordica charantia*, sterols and flavinoids in *Butea monosperma*, and tannins in *Syzygium cuminii* all showed a potent antidiabetic activity (*Ghoshal & Saoji, 2013*). In addition, in Slovakia, *Apis mellifera* BP dominant in rape *Brassica napus* floral pollen had a higher antioxidative activity than BP dominant in poppy *Papaver somniferum* L. and sunflower *Helianthus annuus* L. floral pollen, respectively, *Fatrcova-Sramkova et al. (2013)*.

The BP collected by the stingless bee, *Melipona fasciculata*, was harvested from three cities in Brazil and revealed different anti-inflammatory and antinociceptive activities, with the highest activities in the BP from Chapadinha City, while the BP collected by

*M. fasciculata* showed higher anti-inflammatory and antinociceptive activities than that collected by *A. mellifera* (*Lopes et al., 2019*).

The BPs collected from different geographical regions in Brazil had different chemical compositions and bioactivities (*Araujo et al., 2017*). Monofloral BPs of *Eucalyptus* spp. and multifloral BP exhibited potent inhibitory activities against $\alpha$-amylase, acetylcholinesterase (AChE), tyrosinase, lipoxygenase, lipase, and hyaluronidase, but with different half maximal inhibitory concentration ($IC_{50}$) values. Monofloral BP of *Cocos nucifera* and *Miconia* spp. also exhibited antioxidant properties.

From 18 samples of mono- and poly-floral BPs harvested from 16 different localities in South Korea, all were found to have anti-oxidant, anti-human $\beta$-amyloid precursor cleavage enzyme, AChE inhibitory (AChEI), anti-human intestinal bacteria, and anticancer activities, but with different $IC_{50}$ values (*Zou et al., 2020*).

In terms of their nutrient compositions, polyfloral BPs gave more benefits than mono-floral ones, in terms of having more diverse secondary metabolite-like compounds. Metabolomics analysis revealed that beehive pollen from diverse species of plants contained key ingredients for health (lactate, a pentose sugar, myo-inositol, phosphate, and a furanose), but not in BP dominated by canola floral pollen (*Arathi, Bjostad & Bernklau, 2018*). In addition, *A. mellifera* fed with polyfloral BP were more tolerant to the microsporidian *Nosema ceranae* (*Di Pasquale et al., 2013*). However, in terms of the quality control standards for BP consumption by humans and its industrial production, monofloral BP is more fruitful because known bioactive molecules and their precise concentration can be more easily and economically ascertained (*Kostic et al., 2021*).

The BPs from several countries, including Argentina, Brazil, China, and Spain, have been commercialized after their chemical compositions and bioactivities were reported (*European Union, 2006*). However, little is known about the BP in Thailand, which is high in both bee and plant diversities (*Chantarudee et al., 2012*; *Rattanawannee & Chanchao, 2011*).

In order to identify the bioactive molecules in BP that originated from native plants in Thailand, monofloral BP harvested by *A. mellifera* from six types of floral pollen was examined. The type of floral pollen in each of the BPs (BP1 to BP6) were first identified by palynological analysis using light microscopy (LM) and scanning electron microscopy (SEM), and then by molecular analysis examining the partial sequence of the second internal transcribed sequence (ITS-2) of the ribosomal RNA genes. Then, each of the six BPs was sequentially extracted with methanol, dichloromethane (DCM), and hexane, and the partitioned extracts were screened for $\alpha$-amylase, AChE, and lipase inhibitory activities. Among these bioactivities, the lipase inhibitory activity was targeted due to there being a few previous reports on this activity in BP. The most active sample was subjected to further fractionation by chromatography, and the obtained fractions were analyzed for purity/composition by thin layer chromatography (TLC) and, for fatty acids, by gas chromatography with a flame ionization detector (GC-FID) after conversion to fatty acid methyl esters (FAMEs). Seemingly pure active compounds were analysed by nuclear magnetic resonance (NMR). A chemical compound marker was revealed in the typical bee pollen. The obtained data support the safety and benefit of BP consumption.

## MATERIALS & METHODS

### Chemicals and reagents

The chemicals ($\alpha$-amylase, acarbose, AChE from electric eel, acetylthiocholine iodide, 5,5′-dithiobis (2-nitrobenzoic acid), physostigmine, crude porcine pancreatic lipase type II, p-nitrophenyl palmitate, and orlistate) used in this study were from Sigma-Aldrich, Darmstadt, Germany.

### Sample collection

Six *A. mellifera* BPs (BP1 to BP6), one from each of six localities in Thailand, were collected in 2018, that were suspected, based on the available flowers near the hives, of being monofloral and derived from *Camellia sinensis* L. and *Mimosa diplotricha* in Chiangmai province (BP 1 and BP3, respectively), *Helianthus annuus* L. in Lopburi province (BP2), *Nelumbo nucifera* in Nakhon Sawan province (BP4), *Xyris complanata* in Udon Thani province (BP5), and *Ageratum conyzoides* in Lamphun province (BP6), repectively. The BP samples used were dried using a specific process and stored at room temperature (25 °C) until used.

### Identification of the bee pollen
#### By palynological analysis

The morphology of each BP was initially observed under LM at 400X magnification. Briefly, the respective BP was dispersed in distilled water (d-$H_2O$) on a glass slide, and pictures and characteristics of the BPs were recorded and compared to the reported publications.

Next, the morphology was observed under SEM at 1,500–5,000X magnification. Briefly, BPs were washed three times with ethanol (5 min each), and then three times with acetone (5 min each). The samples were then sent for SEM and energy dispersive X-ray spectrometry (6610LV; Tokyo, Japan) imaging at the Scientific and Technological Research Equipment Center of Chulalongkorn University. The morphology and characteristics of the BPs were recorded and compared to the reported publications.

#### By molecular analysis

Each BP was further identified by comparison of the partial ITS-2 DNA sequences (*Richardson et al., 2015*) to the species-annotated sequences in the GenBank database. For this, the genomic DNA was extracted from approximately 100 mg of BP using a DNeasy Plant Mini Kit (catalog no. 69104; Qiagen, Hilden, Germany). The quality of the extracted DNA was determined by 1.2% (w/v) agarose gel electrophoresis and the ratio of absorbance at 260 and 280 nm. After DNA isolation, the ITS-2 region was amplified using the polymerase chain reaction (PCR) with the forward (5′-ATGCGATACTTG GTGTGAAT-3′) and reverse (5′-GACGCTTCTCCAGACTACAAT-3′) primers. Each PCR amplification was performed in a 25 μL final reaction volume comprised of 12.5 μL of 2X EmeraldAmp® PCR master mix (catalog # RR300A; Takara), 1 μL of each of primer (10 μM), at least 30 ng of genomic DNA template, and nuclease-free d-$H_2O$. The PCR thermal cycling was performed as 98 °C for 30 s, followed by 30 cycles of 98 °C for 10 s, 59 °C for 30 s, and 72 °C for 30 s; and then a final 72 °C for 10 min. The PCR product (500 bp) was checked by 1.2% (w/v) agarose gel electrophoresis in 1X Tris-borate-EDTA buffer at 80

V for 45 min after Ecodye staining. The PCR product was extracted using QIAquick PCR Purification Kit (catalog no. 28106; Qiagen, Hilden, Germany) and sent for commercial direct sequencing. The obtained sequences were used to search for homologous sequences in the GenBank database of the National Center for Biotechnology Information using the Basic Local Alignment Search Tool for nucleotide (BLASTn) algorithm.

## Crude extraction and partition

The extraction and partition was modified from *Chantarudee et al. (2012)*. Each BP (140 g) was mixed with 800 mL of methanol (MeOH), shaken at 100 rpm, 15 °C for 18 h, and then centrifuged at 6,000 rpm, 4 °C, for 15 min. The supernatant was collected, while the solid residue (pellet) was re-extracted three more times in the same manner with 800 mL of MeOH each time. The supernatants were pooled and evaporated under reduced pressure at a maximum temperature of 40–45 °C to obtain the crude MeOH extracts, which were kept at −20 °C in the dark until used.

The MeOH crude extracts were sequentially partitioned by hexane (low polarity), to eliminate the lipid and non-polar compounds; DCM (medium polarity), and finally MeOH (high polarity). To this end, the six MeOH crude extracts (one for each of BP1–6) were separately dissolved in MeOH until it was not sticky and then mixed with an equal volume of hexane in a separating funnel and left to phase separate, whereupon the upper hexane phase was collected. The lower MeOH phase was then further extracted with hexane in the same manner twice more, and the hexane extracts were pooled and evaporated under reduced pressure at a maximum temperature of 40–45 °C to yield the hexane partitioned (HX) extracts of BP1–6 (bee pollen from *C. sinensis* L., *H. annuus* L., *M. diplotricha*, *N. nucifera*, *X. complanata*, and *A. conyzoides,* respectively, and designated as HXCBP, HXHBP, HXMBP, HXNBP, HXXBP, and HXABP, respectively). Meanwhile, the residual MeOH phase was then extracted with an equal volume of DCM three times in the same manner as above (except the DCM phase was the lower layer), with the pooled DCM extracts evaporated as above. The sample from this step was named the DCM partitioned extracts of *C. sinensis* L., *H. annuus* L., *M. diplotricha*, *N. nucifera*, *X. complanata* and *A. conyzoides* flower BP (BP1–6, respectively), and designated as DCMCBP, DCMHBP, DCMMBP, DCMNBP, DCMXBP, and DCMABP, respectively.

Finally, the residual MeOH phase was evaporated as above to yield the MeOH-partitioned (MT) extract of *C. sinensis* L., *H. annuus* L., *M. diplotricha*, *N. nucifera*, *X. complanata* and *A. conyzoides* flower BPs (BP1–6, respectively, and designated as MTCBP, MTHBP, MTMBP, MTNBP, MTXBP, and MTABP, respectively). All partitioned extracts were kept at −20 °C in the dark until used to test the biological activities.

## *In vitro* α-amylase inhibitory activity

The α-amylase inhibition assay was modified from *Akoro, Ogundare & Oladipupo (2017)*. The partitioned extract of BP was dissolved in dimethyl sulfoxide (DMSO) and subsequently diluted in MeOH at different concentrations (125, 250, 500, 1,000, and 2,000 µg/mL). Two hundred and fifty µL of α-amylase solution [0.5 units (U)/mL] dissolved in buffer [$Na_2HPO_4$/$NaH_2PO_4$ (0.02 M) and NaCl (0.006 M)] at pH 6.9 was mixed with 250 µL of

the extract and incubated at 37 °C for 10 min. After that, 250 µL of the starch solution [0.5% (w/v) in d-H$_2$O] was added and incubated at 37 °C for 10 min. The reaction was terminated by the addition of 500 µL DNSA reagent (12 g of sodium potassium tartrate tetrahydrate in 8.0 mL of 2 M NaOH, and 20 mL of 96 mM of 3,5-dinitrosalicylic acid solution) and was heated at 85–90 °C for 5 min in a water bath. The mixture was cooled to room temperature and was diluted with 5 mL of d-H$_2$O, and the absorbance was measured at 540 nm (A$_{540}$) using a UV-Visible spectrophotometer (Sunrise, Tecan, Austria). Acarbose was used as the positive inhibitor. Each sample was performed and measured in triplicate. The inhibitory percentage of $\alpha$-amylase was calculated using the equation given below.

Percentage of $\alpha$-amylase inhibition = [{(A-B) −(C-D)} / (A-B)] ×100

where A is the A$_{540}$ after incubation without an extract, B is the A$_{540}$ after incubation without an extract and $\alpha$-amylase, C is the A$_{540}$ after incubation with an extract and $\alpha$-amylase, and D is the A$_{540}$ after incubation with an extract, but without $\alpha$-amylase.

The % $\alpha$-amylase inhibition (Y axis) was plotted against the extract concentrations (X axis) and the IC$_{50}$ value was obtained using regression analysis.

### *In vitro* AChEI inhibitory activity

Evaluation of the AChEI activity was modified from *Li et al. (2019)* based on Ellman's method. Firstly, 160 µL of TTB [50 mM Tris–HCl buffer pH 8 with 1% (v/v) Trition X-100], 20 µL of the extract dissolved in DMSO (500 µg/mL), and 10 µL of 0.2 U/mL AChE from electric eel dissolved in 0.1% (w/v) bovine serum albumen in TTB were mixed and incubated at 4 °C for 20 min. Then, 5 µL of 15 mM acetylthiocholine iodide in d-H$_2$O and 5 µL of 2 mM 5,5′-dithiobis (2-nitrobenzoic acid) (DTNB) in TTB containing 0.1 M NaCl and 0.02 M MgCl$_2$ were added per well and incubated at 37 °C for 20 min. The absorbance at a wavelength of 412 nm (A$_{412}$) was measured using a microplate reader (Sunrise, Tecan, Austria). Physostigmine was used as the positive control. All the reactions were performed in triplicate. The percentage inhibition was calculated as follows.

Percentage of AChE inhibition = [{(A-B) −(C-D)} / (A-B)] ×100

where A is the A$_{412}$ after incubation without an extract, B is the A$_{412}$ after incubation without an extract and AChE, C is the A$_{412}$ after incubation with an extract and AChE, and D is the A$_{412}$ after incubation with an extract, but without AChE.

The % AChEI (Y axis) was plotted against the extract concentrations and the IC$_{50}$ values were obtained using linear or non-linear regression analysis.

### *In vitro* porcine pancreatic lipase inhibitory (PPLI) activity

The enzyme solution was prepared immediately before use as previously described (*Jamous et al., 2018*) with some modifications. Crude PPL type II was suspended in 50 mM Tris–HCl buffer pH 8 to a concentration of 2 mg/mL. The suspension was mixed and centrifuged at 16,000x g for 10 min. The clear supernatant was recovered and kept. The PPLI assay was adapted from *Maqsood et al. (2017)*. Briefly, 100 µL of the extract at different concentrations (200, 400, 600, 800, and 1,000 µg/mL for DCMCBP, DCMMBP, DCMNBP, DCMXBP, and DCMABP; and 12.5, 25, 50, 100, and 200 µg/mL for DCMMBP2 and DCMMBP2-1) dissolved in DMSO and 600 µL of 50 mM Tris–HCl buffer (pH 8.0) were pre-incubated

with 200 µL of 2 mg/mL of PPL solution at 37 °C for 30 min. Afterwards, 100 µL of 1.5 mM of p-nitrophenyl palmitate (p-NPP) in isopropanol was added and incubated at 37 °C for 2 h. Lipase activity was determined by measuring the hydrolysis of p-NPP to p-nitrophenol product *via* measuring the absorbance at 410 nm ($A_{410}$) using a microplate reader. Orlistat was used as the positive standard. Each sample was performed and measured in triplicate. The percentage of lipase inhibition was calculated according the following formula.

Percentage of lipase inhibition $= [\{(A-B) -(C-D)\} / (A-B)] \times 100$

where A is the $A_{410}$ after incubation without an extract, B is the $A_{410}$ after incubation without an extract and lipase, C is the $A_{410}$ after incubation with an extract and lipase, and D is the $A_{410}$ after incubation with an extract, but without lipase.

The % lipase inhibition (Y axis) was plotted against the extract concentrations (X axis) and the $IC_{50}$ values were obtained from the graph.

### Enrichment of active fractions

Among the mentioned bioactivities, the extract with the most potent *in vitro* PPLI activity was used for further fractionation (enrichment) by silica gel 60 column chromatography (SiG60-CC).

### *(A) Large scale SiG60-CC (500-mL column)*

A 500-mL column was packed with fine SiG60 (Merck). The partitioned extract (6.0 g) was dissolved in 20 mL of MeOH and combined with 20 g of rough SiG60 and allowed to dry, whereupon it was poured over the surface of the packed SiG60 column. The column was first eluted with 6.5 L of DCM, followed by 8.5 L of 7% (v/v) MeOH in DCM and then 3.5 L of MeOH, respectively. Eluted fractions (250 mL each) were collected, and the solvent was removed by evaporation under reduced pressure at a maximum temperature of 40–45 °C. The pattern of chemical compounds in each fraction was profiled by TLC (see below). Fractions with the same TLC pattern were pooled together and tested for *in vitro* PPLI activity using the assay as above.

### *(B) Small scale SiG60-CC (250-mL column)*

A 250-mL column was packed with fine SiG60. The active fraction (300 mg) was dissolved in five mL of MeOH and combined with 5 g of rough SiG60 and allowed to dry, whereupon it was poured over the surface of the packed SiG60 column. The column was eluted with 1,000 mL of 2% (v/v) MeOH in DCM and then 500 mL of MeOH, respectively. Eluted fractions (seven mL each) were collected, and the solvent was removed by evaporation under reduced pressure at a maximum temperature of 40–45 °C. The pattern of chemical compounds in each fraction was profiled by TLC (see below). Fractions with the same TLC pattern were assumed to be chemically similar and were pooled. After that, each fraction was tested for its *in vitro* PPLI activity using the assay as above.

### *One-dimensional TLC*

A 5 ×5 cm² TLC plate with silica as the immobile phase was prepared. The sample was spotted onto the solvent front line of the plate by a capillary tube, allowed to dry at room temperature, and then resolved in one direction using the appropriate mobile phase solvent
of 7% (v/v) MeOH: DCM. The resolved compounds on the TLC plate were visualized under UV light at 254 nm or by dipping in 3% (v/v) anisaldehyde in MeOH and heating over a hot plate.

## Chemical structure analysis by NMR

Among the fractions obtained from the SiG60-CC (250-mL size), the most active fraction for PPLI activity was evaporated and analysed. Briefly, the evaporated sample was dissolved in an appropriate deuterated solvent (Chloroform-d or MeOH-d4, Merck) at a ratio of 5–20 mg of compound to 600 $\mu$L of deuterated solvent. It was then transferred to an NMR tube and shaken until completely dissolved. The NMR spectrum was recorded on a Jeol JNM-ECZ (JNM-ECZ500R, Tokyo, Japan) 500MHz operated at 500 MHz for [1]H-NMR nuclei in order to detect the functional groups using tetramethylsilane as the internal standard. The chemical shift in $\delta$ (ppm) was assigned with reference to the signal from the residual protons in the deuterated solvents, while the chemical shift and J coupling value were determined using the MestReNova version 12.0.3 software.

## Preparation of FAMEs

A portion (26.7 mg) of the fraction with the highest PPLI activity obtained from SiG60-CC (250-mL size), was added to absolute MeOH (two mL), followed by 0.5 mL of concentrated $H_2SO_4$. The reaction mixture was stirred and heated at 60 °C for 3.5 h. The reaction mixture was then evaporated to dryness, diluted with DCM (six mL) and extracted with saturated $NaHCO_3$. The DCM layer was collected and washed several times with d-$H_2O$ until pH of the solution was 7. The combined DCM layer was dried over anhydrous $Na_2SO_4$ and evaporated to dryness. Prior to analysis of the prepared FAMEs by GC-FID analysis (6890N GC, California, USA), the formation of FAMEs was confirmed by [1]H-NMR analysis (JNM-ECZ500R, Tokyo, Japan), where the singlet signal of the methoxy group at $\delta_H$ 3.65 ppm was observed (Supplement 3).

## GC-FID analysis

The prepared FAMEs of the fraction with the highest PPLI activity from the SiG60-CC (250-mL size) fractionation was submitted to the Food Research and Laboratory (Faculty of Science, Chulalongkorn University) for analysis of its fatty acid components by GC-FID following the AOAC method 996.06. Briefly, chromatographic analysis was performed using a GC-FID system (Aglilent 6890 N) equipped with an autosampler and split-splitless injector. A SPTM 2560 FULED SILICA capillary column with an internal diameter of 0.25 mm and 0.2 $\mu$m film thickness was used for the chromatographic separation. Helium was used as the carrier gas at 1.1 mL/min. The injector and detector temperatures were set at 260 and 250 °C, respectively. The initial GC oven temperature was 140 °C, held for 5 min, increased to 240 °C at 20 °C/min, and held at 250 °C for 0.5 min. A volume of 1.0 $\mu$L of sample was injected using the split injection mode (100:1). The peaks were identified by comparison of their relative retention times with a standard FAME mixture. The results were expressed as mg/g fatty acid.

### Data analysis

Experiments were performed in triplicate. Numerical data are reported as the mean $\pm$ one standard deviation ($\pm$ SD), determined in the Microsoft Excel 2019 software. One-way ANOVA and $T$-test were used to test for significant differences in $IC_{50}$ values. Tukey's test ($p < 0.05$) was applied for the pairwise multiple comparisons. The statistical analyses were performed using IBM SPSS statistics version 22 for windows.

The overall procedure of BP screening and enrichment for the PPLI active component in the most active extract is summarized schematically in Fig. 1.

## RESULTS

### Palynological analysis of the six BP samples

The major floral pollen types present in BP must be identified in order to report the type of floral pollen present in the BP. The simplest way is by morphology using LM and SEM analyses. Under LM and SEM, the morphology of bee pollen was observed. The pollen grains in BP1 were convex triangular with tricolporate, and the exine ornamentation was verrucate (Figs. 2A and 3A). This is consistent with the pollen grains of *C. sinensis* (L.) Kuntze (*Ariyarathna et al., 2011*; *Fan et al., 2019*). The pollen grains of BP2 were close to spherical with triaperture, and the exine ornamentation was echinate (Figs. 2B and 3B), supported it to be the pollen grains of *H. annuus* L. (*Ali et al., 2021*). The pollen grains of BP3 were spherical or prolate spheroidal in shape with four (tetrad) pollen subunits, and the exine ornamentation was tuberculate (Figs. 2C and 3C), supporting it to be the pollen grains of *M. diplotricha* (*Lima, Silva & Santos, 2008*; *Peukpiboon, Benbow & Suwannapong, 2017*). The pollen grains of BP4 were spherical in shape with tricolpate, and the exine ornamentation was uniformly dense reticulate (Figs. 2D and 3D), supporting that they were the pollen grains of *N. nucifera* (*Sangsuk, Baslev & Jampeetong, 2021*; *Zhang et al., 2019*). The pollen grains of BP5 were ellipsoidal in polar view, flattened/convex in equatorial view with monosulcate and operculate, and the exine ornamentation was reticulate (Figs. 2E and 3E), they consitent with the pollen grains of *X. complanata* (*Da Luz et al., 2015*). Finally, the pollen grains of BP6 were spherical in shape with tricolporate, and the exine ornamentation was echinate (Figs. 2F and 3F) confirming they were the pollen grains of *A. conyzoides* (*Zafar, Ahmad & Khan, 2007*; *Garg, 1996*).

### Identification of the pollen species in each BP sample by molecular analysis

The six BP samples (BP1–6) were also identified by sequence analysis of the ITS-2 region of the rRNA genes. In each case, PCR amplification of the ITS-2 region gave an amplicon of the expected size (500 bp). After sequencing the amplicons and using them as the query sequence to BLASTn search the GenBank reference sequences, the same plant pollen species identifications were obtained as by the LM and SEM morphological analyses. Sample BP1 showed 100% nucleotide identity to the sequence of *C. sinensis* L. (accession # MN242039.1), BP2 at 95.33% nucleotide identity to the sequence of *H. annuus* L. (accession # KF767534.1), BP3–6 at 100% nucleotide identity to the sequence of *M.*

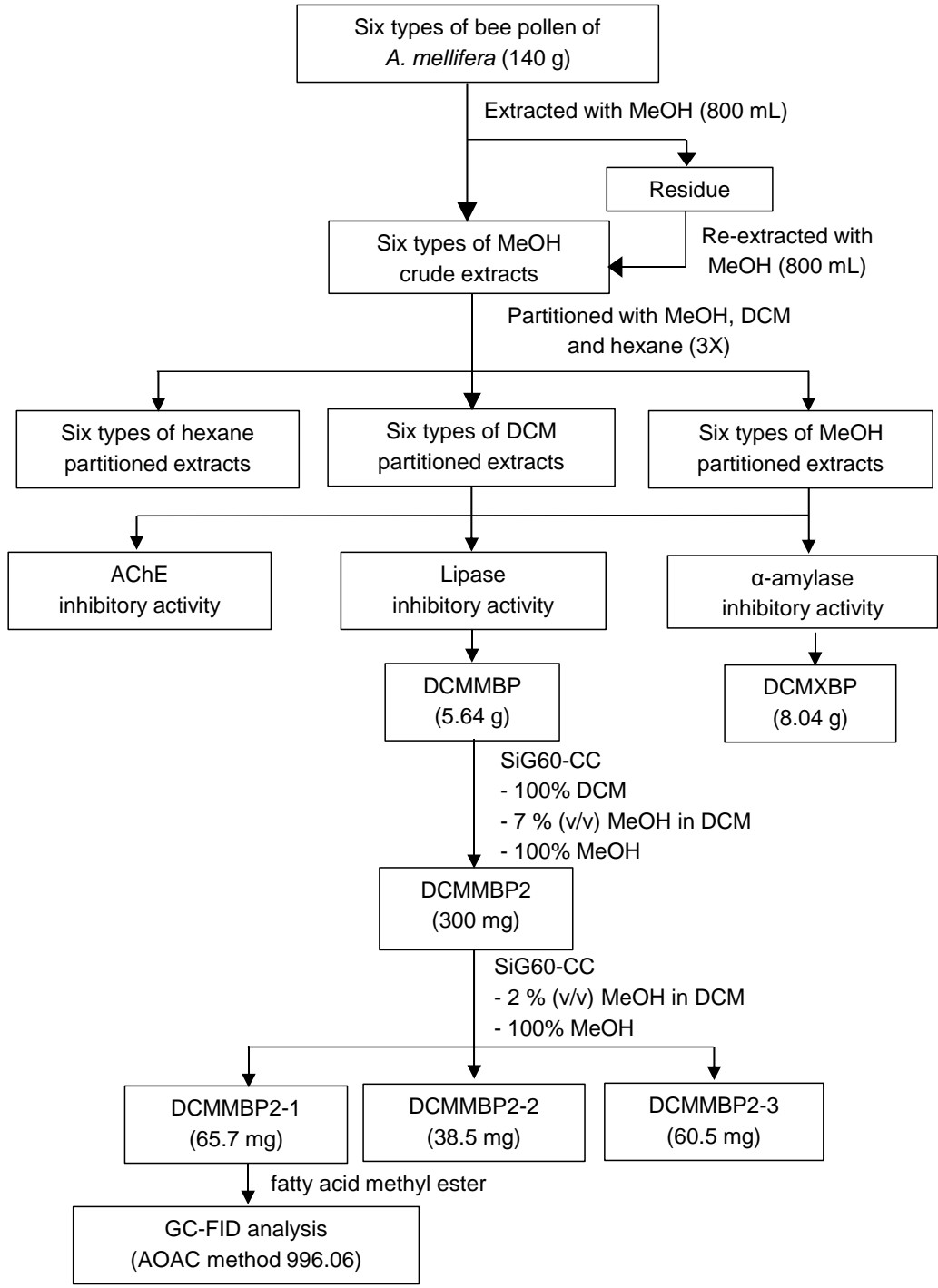

**Figure 1** Summary for extraction, screening and enrichment procedures for the selected BP.

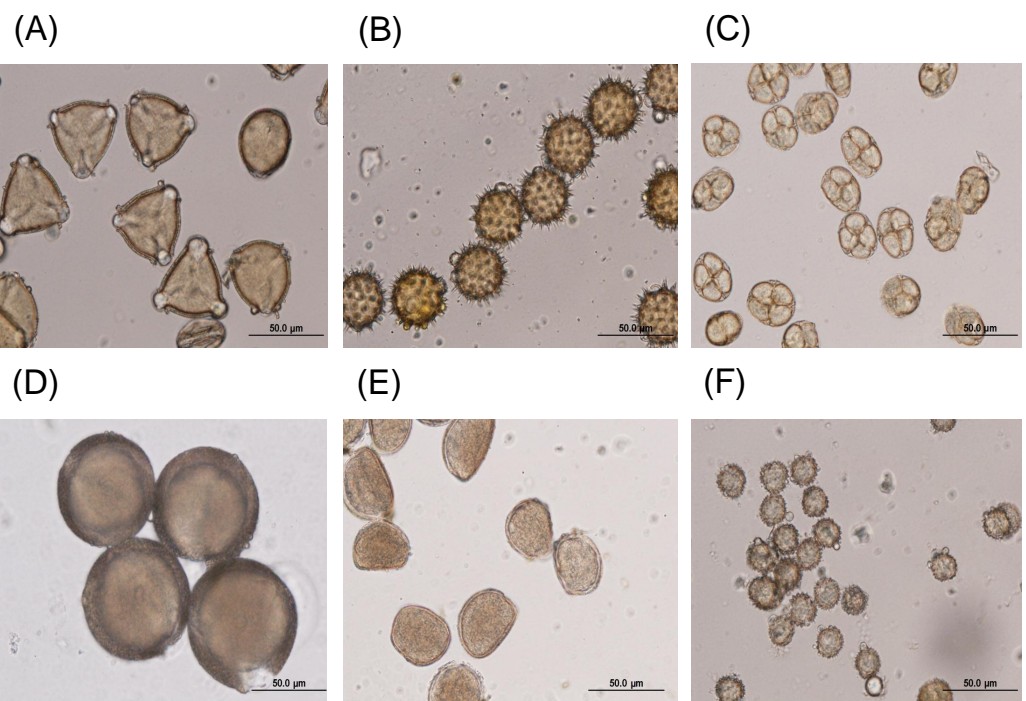

**Figure 2** Representative LM images of (A–F) BP1–6, respectively, identified as (A) *C. sinensis* L., (B) *H. annuus* L., (C) *M. diplotricha*, (D) *N. nucifera*, (E) *X. complanata*, and (F) *A. conyzoides* flower pollen.

*diplotricha* (accession # MH768249.1), *N. nucifera* (accession # FJ599761.1), *X. complanata* (accession # MW113223), and *A. conyzoides* (accession # KY700213.1), respectively.

Thus, the six BPs were confirmed to be essentially monofloral, matching the principal flowers around the hives, and with the morphological and molecular analyses congruent with each other. Thus, floral origin of each BP was clarified in this work.

## The partitioned extracts of BP

For the six BPs, they were separately sequentially partitioned by MeOH, DCM, and hexane, three organic solvents with different polarities. The yield and character of all 18 obtained extracts (three solvents for each of BP1-6) are summarized in Table 1. The highest yield was obtained from the MeOH-partitioned extracts in all six samples (above 40%). Only a sticky solid form was obtained for the DCM-partitioned extracts, while an oil form was obtained in both the MeOH- and hexane-partitioned extracts.

However, only the MeOH and DCM partitioned extracts of all six types of BP were tested for the three enzyme inhibitory activities because the hexane-partitioned extracts were insoluble in each of the respective enzyme assay buffer solutions.

### *In vitro* α-amylase inhibitory activity

The partitioned extracts were initially used at a final concentration of 2 mg/mL, and the α-amylase inhibitory activity (%) is presented as the mean ± SD in Table 2. At this concentration, DCMXBP provided the highest *in vitro* α-amylase inhibitory activity

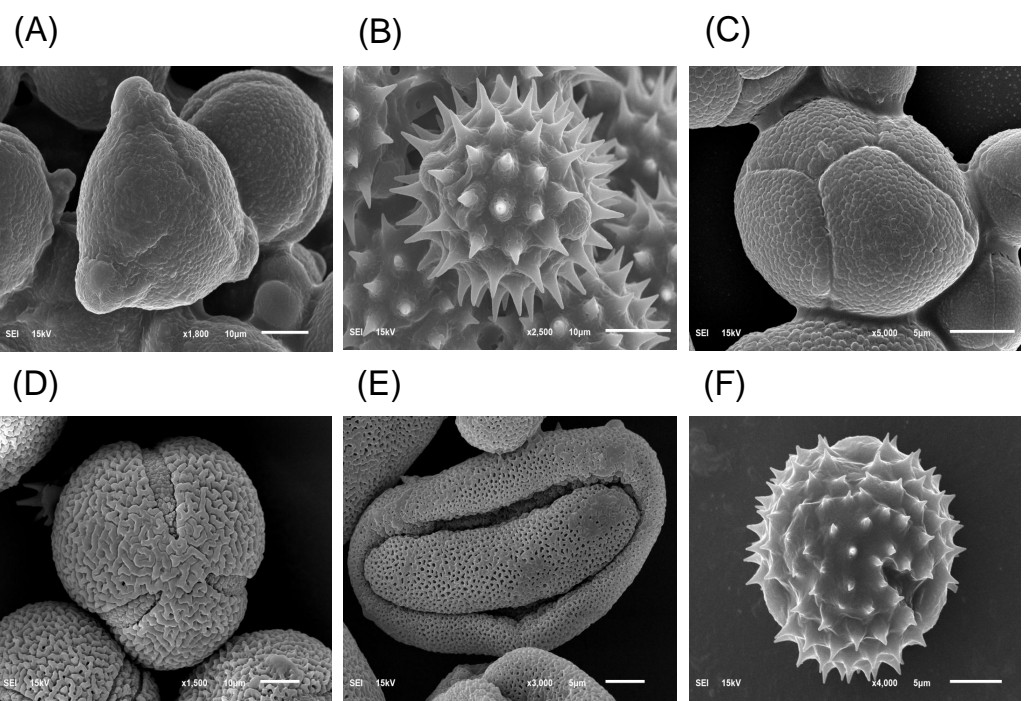

**Figure 3** Representative SEM images of (A–F) BP1–6, respectively, identified as (A) *C. sinensis* L., (B) *H. annuus* L., (C) *M. diplotricha*, (D) *N. nucifera*, (E) *X. complanata*, and (F) *A. conyzoides* flower pollen.

(54.8 ± 2.8%). The subsequent dose response assay revealed the anti $\alpha$-amylase activity of DCMXBP was concentration dependent (Fig. 4A, Supplement 1) with an $IC_{50}$ value of 1,792.5 ± 51.0 $\mu$g/mL (Table 2), which was markedly less effective than that of acarbose, the positive control (Fig. 5A, Supplement 2) with a 63-fold lower $IC_{50}$ value of 28.1 ± 2.7 $\mu$g/mL. In contrast, DCMMBP and MTXBP had no real anti-$\alpha$-amylase activity at this concentration of 2 mg/mL (1.19 ± 2.06% and 0.00 ± 0.00%, respectively).

## In vitro AChEI activity

The partitioned extracts were initially screened for AChEI activity at a final concentration of 500 $\mu$g/mL. The *in vitro* AChEI activity (%) is presented as the mean ± SD in Table 2. At this concentration, DCMCBP provided the highest AChEI activity at 19.3 ± 1.5%. However, the AChEI activity of all the partitioned extracts were negligible compared to that of physostigmine, the positive control, with an over 230-fold lower $IC_{50}$ value of 0.082 ± 0.002 $\mu$g/mL (Fig. 5B and Table 2, Supplement 2).

## *In vitro* PPLI activity

The partitioned extracts were initially screened for PPLI activity at a final concentration of 400 $\mu$g/mL with the results (as the % PPLI activity) presented as the mean ± SD in Table 2. This concentration was selected because it was the highest concentration that could be totally dissolved in DMSO. At this concentration, some DCM partitioned extracts showed a PPLI activity close to 50%, and so they were further evaluated at different

**Table 1  The weight, yield, and character of the partitioned extracts.**

| Sample | Weight (g) | Yield (%) | Character |
|---|---|---|---|
| MTCBP | 64.93 | 46.38 | Pale brown oil |
| DCMCBP | 7.41 | 5.29 | Sticky dark brown solid |
| HXCBP | 7.73 | 5.52 | Dark brown oil |
| MTHBP | 79.97 | 57.12 | Dark brown oil |
| DCMHBP | 12.55 | 8.96 | Sticky dark brown solid |
| HXHBP | 8.98 | 6.41 | Dark brown oil |
| MTMBP | 57.90 | 41.36 | Dark brown oil |
| DCMMBP | 9.89 | 7.06 | Sticky dark brown solid |
| HXMBP | 7.43 | 5.31 | Dark brown oil |
| MTNBP | 79.49 | 56.8 | Pale brown oil |
| DCMNBP | 4.68 | 3.34 | Sticky brown solid |
| HXNBP | 5.31 | 3.79 | Pale brown oil |
| MTXBP | 96.81 | 69.15 | Dark brown oil |
| DCMXBP | 8.04 | 56.8 | Sticky dark brown solid |
| HXXBP | 11.7 | 8.36 | Dark brown oil |
| MTABP | 79.28 | 56.63 | Dark brown oil |
| DCMABP | 5.75 | 4.11 | Sticky dark brown solid |
| HXABP | 8.47 | 6.05 | Dark brown oil |

**Table 2  The percentage of enzyme inhibition (mean ±S.D.) and IC50 value ($\mu$g/mL) of the partitioned extracts.**

| Sample | $\alpha$-amylase / $IC_{50}$ | AChE/$IC_{50}$ | PPLI/$IC_{50}$ |
|---|---|---|---|
| MTCBP | 16.12 ± 1.81 / - | 7.74 ± 0.23 / - | 7.74 ± 0.56 / - |
| DCMCBP | 38.58 ± 3.87 / - | 19.25 ± 1.50 / - | 49.47 ± 1.31 / 458.48 ± 13.38[b] |
| MTHBP | 14.31 ± 0.99 / - | 11.34 ± 0.92 / - | 29.86 ± 1.15 / - |
| DCMHBP | 42.70 ± 2.49 / - | 8.23 ± 1.97 / - | 3.65 ± 1.72 / - |
| MTMBP | 11.68 ± 1.05 / - | 13.95 ± 1.51 / - | 14.38 ± 1.38 / - |
| DCMMBP | 1.19 ± 2.06 / - | 10.22 ± 3.72 / - | 42.89 ± 2.23 / 500.80 ± 24.76[b] |
| MTNBP | 7.92 ± 1.12 / - | 4.13 ± 0.47 / - | 31.58 ± 0.64 / - |
| DCMNBP | 4.50 ± 2.88 / - | 5.91 ± 1.74 / - | 39.49 ± 1.48 / 876.09 ± 24.15[d] |
| MTXBP | 0.00 ± 0.00 / - | 7.37 ± 1.71 / - | 38.51 ± 0.13 / - |
| DCMXBP | 54.82 ± 2.76 / 1,792.48 ± 50.96[b] | 7.76 ± 3.67 / - | 35.09 ± 1.04 / 960.49 ± 38.19[e] |
| MTABP | 13.72 ± 1.44 / - | 6.81 ± 0.85 / - | 14.53 ± 0.94 / - |
| DCMABP | 41.45 ± 4.23 / - | 9.95 ± 3.59 / - | 37.38 ± 2.39 / 646.09 ± 19.41[c] |
| Acarbose | - / 28.08 ± 2.65[a] | – | – |
| Physostigmine | | - / 0.082 ± 0.002 | – |
| Orlistat | – | – | - / 0.021 ± 0.000[a] |

**Notes.**
The $IC_{50}$ values were calculated using nonlinear regression except for DCMNBP and orlistat that were calculated using linear regression. Data are shown as the mean. Within a column, means with a different superscript letter are significantly different [$p < 0.05$; One-way ANOVA for anti-amylase activity ($p = 0.000$) and Post Hoc (Tukey) test for anti-lipase activity ($p = 0.000$ except $p$ between DCMNBP and DCMXBP $= 0.008$].

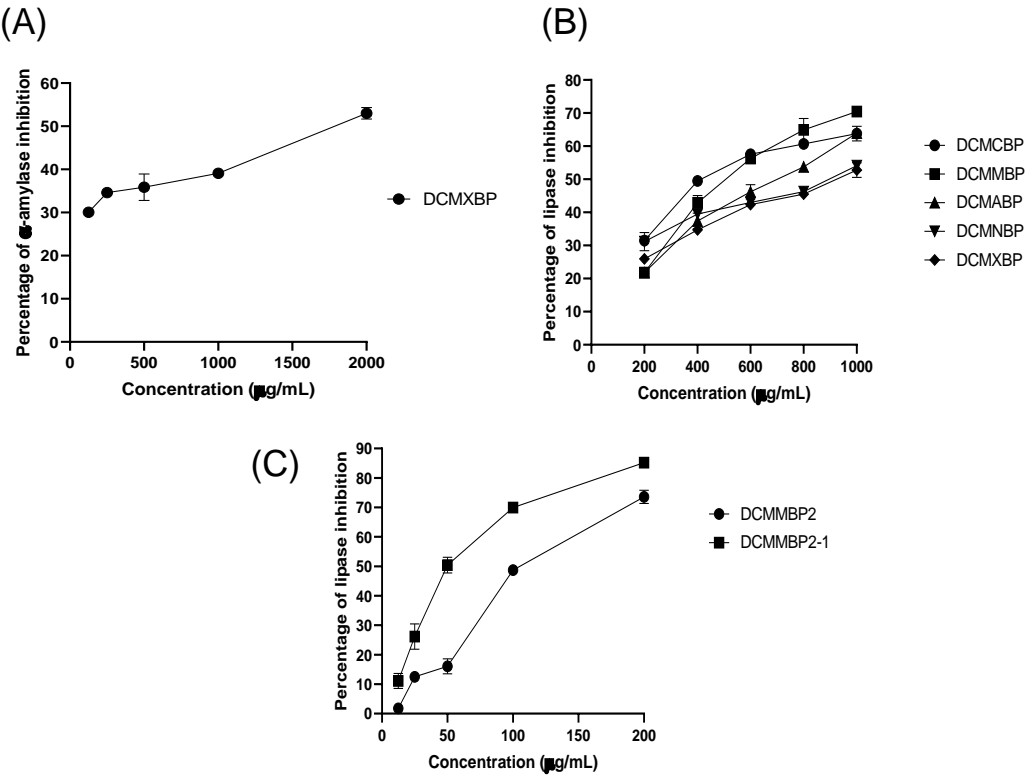

**Figure 4** The (A) α-amylase inhibition activity (%) of DCMXBP, (B) PPLI activity (%) of DCM partitioned extracts, and (C) PPLI activity (%) of DCMMBP2 and DCMMBP2-1. Data are shown as the mean ± SD.

concentrations. The PPLI activity was found to be dose-dependent (Fig. 4B, Supplement 1) and broadly similar between DCMCBP and DCMMBP, with $IC_{50}$ values of 458.5 ± 13.4 and 500.8 ± 24.8 μg/mL, respectively. However, they were markedly less effective than orlistat, the positive control, with an over 21,800-fold lower $IC_{50}$ value of 0.021 ± 0.000 μg/mL (Fig. 5C, Supplement 2).

### *In vitro* PPLI activity of active compounds from DCMMBP
### Fractionation of DCMMBP by SiG60-CC

From Table 2, although the DCMCBP and DCMMBP had no marked difference in their PPLI activity ($IC_{50}$ of 458.5 ± 13.4 and 500.8 ± 24.8 μg/mL, respectively), DCMMBP was selected for further fractionation by SiG60-CC because there have been numerous previous studies on the lipase inhibitory activity of *Camellia sinensis* L (*Chen et al., 2020*), but the lipase inhibitory activity of *M. diplotricha* flower BP has not been reported yet.

From 5.64 g of DCMMBP, a total of 74 fractions were collected. After comparison of their TLC profiles and pooling fractions with a similar pattern, five different fractions (DCMMBP1-5) were obtained. Their weight, yield, and appearance are summarized in Table 3. All five pooled fractions were a sticky solid. Fraction DCMMBP5 provided the highest yield (42.20%). These five pooled fractions were then tested for their PPLI

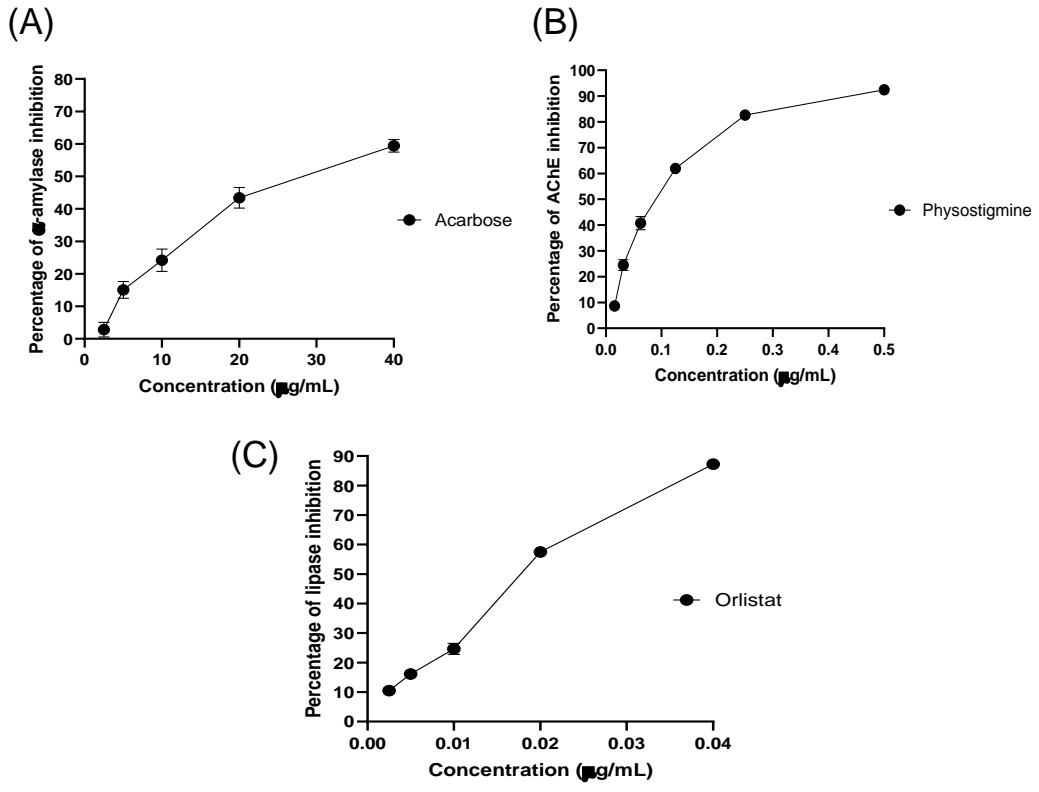

**Figure 5** The (A) α-amylase inhibition (%) of acarbose, (B) AChEI activity (%) of physostigmine, and (C) PPLI activity (%) of orlistat. Data are shown as the mean ± SD.

activity, with the results shown in Fig. 4C and the derived $IC_{50}$ values reported in Table 3 (Supplement 1). Fraction DCMMBP2 gave the highest PPLI activity ($IC_{50}$ of 128.5 ± 3.0 μg/mL).

Since the DCMMBP2 fraction had the highest PPLI activity, it was further enriched by SiG60-CC (250-mL size) to yield a total of 150 fractions. After a pooling of fractions with similar TLC profiles, three different fractions (DCMMBP2-1, DCMMBP2-2, and DCMMBP2-3) were obtained. Their weight, yield, and appearance are summarized in Table 3. With respect to their PPLI activity (Fig. 4C), fraction DCMMBP2-1 had the highest PPLI activity ($IC_{50}$ of 52.6 ± 3.5 μg/ mL), while DCMMBP2-2 and DCMMBP2-3 had essentially no PPLI activity.

## Principal chemical composition analysis (TLC and NMR) of fractions DCMMBP2-1 and DCMMBP2-2

After SiG60-CC, the chemical composition of the three obtained fractions was tested by TLC (Fig. 6). For DCMMBP2-1, no band was observed under UV light at 254 nm (Fig. 6A), but a dark blue spot was found after dipping in 3% (v/v) anisaldehyde in MeOH and heating over a hot plate (Fig. 6B). The structure of fraction DCMMBP 2-1 was first analyzed by $^1$H-NMR, where the signal at δ 0.85 to 2.85 ppm and the signal at δ 5.35

**Table 3  Characteristics and PLLI activity (IC$_{50}$ value) of all pooled fractions after the first (500-mL column) and second (250-mL colume) SiG60-CC fractionation.**

| Sample | Weight (mg) | Yield (%) | Appearance | IC$_{50}$ ($\mu$g/mL) |
|---|---|---|---|---|
| *After 1$^{st}$ SiG60-CC:* | | | | |
| DCMMBP1 | 210 | 3.72 | Sticky brown solid | – |
| DCMMBP2 | 300 | 5.32 | Sticky brown solid | 128.48 ± 3.01[c] |
| DCMMBP3 | 2,060 | 36.52 | Sticky pale-yellow solid | – |
| DCMMBP4 | 410 | 7.27 | Sticky brown solid | – |
| DCMMBP5 | 2,380 | 42.20 | Sticky dark brown solid | – |
| Orlistat | – | – | | 0.021 ± 0.00[a] |
| *After 2$^{nd}$ SiG60-CC:* | | | | |
| DCMMBP2-1 | 65.7 | 21.9 | Sticky brown solid | 52.63 ± 3.50[b] |
| DCMMBP2-2 | 38.5 | 12.83 | Yellow solid | – |
| DCMMBP2-3 | 60.5 | 20.17 | Brown solid | – |
| Orlistat | – | – | | 0.021 ± 0.00[a] |

**Notes.**
Remark: The IC$_{50}$ values of orlistat and DCMXBP2 were calculated from a linear regression, while that for DCMMBP2-1 was calculated from non-linear regression. Values are shown as the mean. Means with a different superscript letter are significantly different ($p = 0.000$ between DCMMBP2 and orlistat and $p = 0.001$ between DCMMBP2-1 and orlistat; Independent-Samples T-Test). DCMMBP2-2 was initially called compound **1** and later identified as naringenin.

ppm showed the characteristics of unsaturated free fatty acids (UFFAs; Supplement 4). Therefore, fraction DCMMBP2-1 was prepared as FAMEs for the GC-FID analysis. In contrast, only a sharp band was observed on the TLC plate for fraction DCMMBP2-2, which indicated it was enriched to apparent homogeneity (potentially pure) compound. Therefore, the chemical structure of the compound, named compound **1**, in fraction DCMMBP2-2 was analyzed by $^1$H-NMR. Compared to naringenin isolated from the *M. pigra* L. flower BP (Khongkarat et al., 2021), the obtained NMR peaks in the chemical shift pattern were $^1$H-NMR (500 MHz, MeOH-d4) $\delta$: 7.30 (d, $J = 8.5$ Hz, 2H), 6.80 (d, 8.5 Hz, 2H), 5.86 (q, $J = 2.2$ Hz, 2H), 5.33 (dd, $J = 13.0$, 3.0 Hz, 1H), 3.09 (dd, $J = 17.1$, 13.0 Hz, 1H), and 2.67 (dd, $J = 17.1$, 3.0 Hz, 1H) (Supplement 5). Thus compound **1** was identified as naringenin (Fig. 7A). However, the structure of DCMMBP2-3 was not identified from the NMR results because it did not have any marked PPLI activity.

### Analysis of the fatty acids of fraction DCMMBP2-1

The fatty acids were identified and divided into saturated fatty acids (SFAs) without C =C double bonds, monounsaturated fatty acids (MUFAs) with one such bond, and polyunsaturated fatty acids (PUFAs) with two or more double bonds between two connected carbon atoms. The fatty acid content in DCMMBP2-1 is summarized in Table 4. The major fatty acids included one SFA (palmitic acid) and two PUFAS (linoleic and linolenic) (Figs. 7B–7D), together with small amounts of stearic acid, oleic acid, myristic acid, pentadecanoic acid, lignoceric acid, margaric acid, ecosadienoic acid, eicosenoic acids, $\gamma$-linolenic acid, behenic acid, palmitoleic acid, heptadecenoic acid, and eicosanoic acid (Supplement 6).

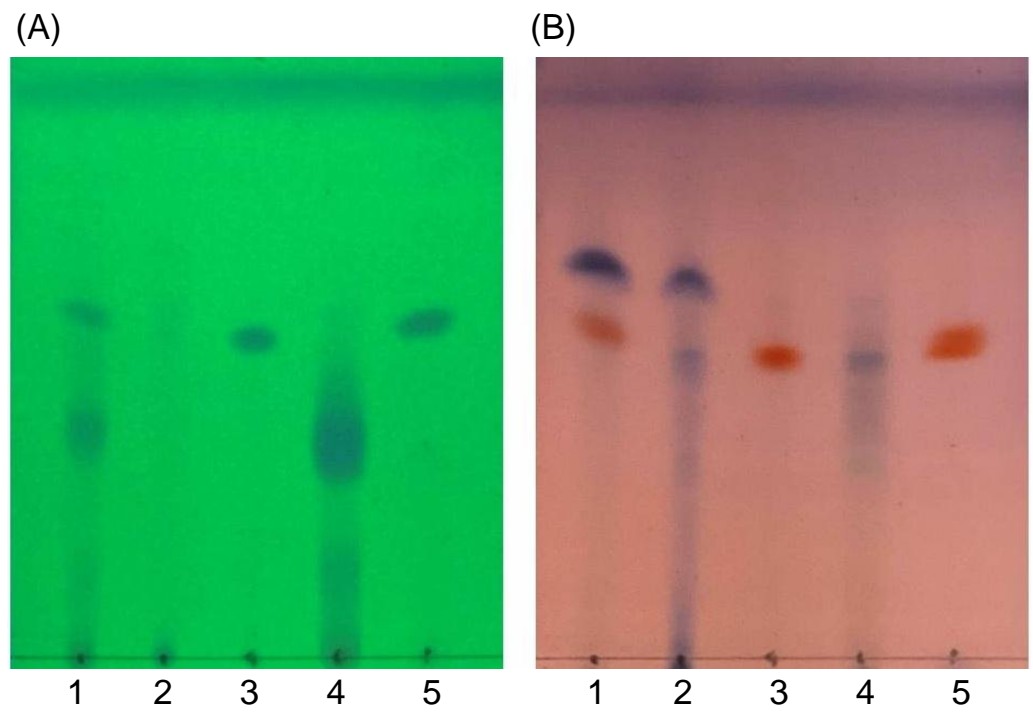

**Figure 6** TLC images showing the profile of DCMMBP2 (lane 1), DCMMBP2-1 (lane 2), DCMMBP2-2 (lane 3), DCMMBP2-3 (lane 4), and naringenin (lane 5) under (A) UV light and (B) after 3% (v/v) anisaldehyde in MeOH.

## DISCUSSION

Aging societies are found in most developed and many developing countries due to the leap in progress in medicine and standard of living (nutrition, shelter, etc.) leading to people living longer. However, these new life styles, especially in the increased access to goods and processed food exposes the population to face many diseases, such as diabetics, Alzeimer's, and obesity. The drugs to treat these diseases are expensive, especially when imported, as is the case in Thailand. These include acarbose and orlistat as expensive anti-diabetic as anti-obesity drugs, respectively. In addition, those drugs can cause adverse side effects. Orlistat can cause several mild-to-moderate gastrointestinal adverse effects, serious adverse hepatic effects, and rare cases of acute kidney injury. In addition, it interferes with the absorption of many drugs, resulting in their decreased bioavailability and effectiveness (*Filippatos et al., 2008*).

Hence, finding alternative treatments from local, sustainable, and natural products is necessary. Many types of fruit, vegetables, and mushrooms (fungi) have been shown to inhibit $\alpha$-amylase activity (*Papoutsis et al., 2021*), while a mixture of flavonoids and phenolics acids from *Aristotelia chilensis* leaves and coumarins have potential AChEI activity (*Cespedes et al., 2017*; *De Souza, Renna & Figueroa-Villar, 2016*). In addition, lipase inhibitors have been reported from both synthetic compounds (phosphonates, boronic acids, fats analogues) and natural compounds from many spices and plants, such as

**Figure 7** Structural formula of naringenin (A) and the chemical structures of palmitic acid, C16:0 (B), linoleic acid, C18:2n6c (C), and α-linolenic acid, C18:3n3 (D).

hydroxybenzoic acids, hydroxycinnamic acid, flavonol, isoflavonoid, flavanone, and hydroxycoumarin from *Momordica charantia* fruits (*Bialecka-Florjanczyk et al., 2018*; *Chanda et al., 2019*; *Sellami et al., 2017*).

In this work, BP harvested by *A. mellifera*, the most well managed honeybee, was our target. Due to the rapid growth of industries in Thailand, many agricultural areas have become fragmented and the agroecosystem has been changed into monocrop cultures, which results in a higher proportion of bee hives having monofloral BP. As known, the bioactivity of BP is mainly dependent on its botanical origin (*Rebiai & Lanez, 2012*). Thus, the monofloral BP from various botanical and geographical origins in Thailand was the focus of this study. With the standard methods used in pollen identification (*Bell et al., 2016*), the six BP samples in this work were ascribed to have originated from *C. sinensis* L., *H. annuus* L., *M. diplotricha*, *N. nucifera*, *X. complanata,* and *A. conyzoides* for BP1–6, respectively. Three bioactivities (anti-amylase, AChEI, and PPLI activities) were focused on because there have been rare reports on these activities in BP, especially in Thailand.

Although *in vitro* assays were used, the obtained data was reliable. The anti-amylase activity was performed using α-amylase from porcine pancreas, which shows a close relationship to human α-amylase (*Butterworth, Warren & Ellis, 2011*). For the AChEI activity, AChE from electric eel, which has the same active site as human AChE, was selected (*Orhan et al., 2011*), while PPL has a similar active site to human pancreatic lipase was used in the anti-lipase assay (*Winkler, d'Arcy & Hunziker, 1990*). Therefore, these

**Table 4  Fatty acid composition of the DCMMBP2-1 (as FAMEs).**

| Peak # | Fatty acid | Type | Abbreviation | Retention time (min) | Fatty acid (mg/g) |
|---|---|---|---|---|---|
| 1 | Myristic acid | SFAs | C14: 0 | 20.187 | 4.4292 |
| 2 | Pentadecanoic acid | SFAs | C15: 0 | 21.877 | 3.1227 |
| 3 | Palmitic acid | SFAs | C16: 0 | 23.586 | 247.7186 |
| 4 | Palmitoleic acid | MUFAs | C16: 1 | 24.766 | 1.6687 |
| 5 | Margaric acid | SFAs | C17: 0 | 25.196 | 2.9375 |
| 6 | Heptadecenoic acid | MUFAs | C17: 1 | 26.224 | 1.3405 |
| 7 | Stearic acid | SFAs | C18: 0 | 26.773 | 18.9883 |
| 8 | Oleic acid | MUFAs | C18: 1n9c | 27.768 | 16.5952 |
| 9 | Linoleic acid | PUFAs | C18: 2n6c | 29.316 | 497.5391 |
| 10 | Eicosanoic acid | SFAs | C20: 0 | 29.799 | 1.0764 |
| 11 | $\gamma$-linolenic acid | PUFAs | C18: 3n6 | 30.379 | 2.2933 |
| 12 | Eicosenoic acids | MUFAs | C20: 1 | 30.717 | 2.5925 |
| 13 | $\alpha$-linolenic acid | PUFAs | C18: 3n3 | 30.930 | 184.2923 |
| 14 | Ecosadienoic acid | PUFAs | C20: 2 | 32.155 | 2.6826 |
| 15 | Behenic acid | SFAs | C22: 0 | 32.663 | 1.8952 |
| 16 | Lignoceric acid | SFAs | C24: 0 | 35.655 | 3.0019 |

*in vitro* assays were used as a screening method to search for potential human enzyme inhibitors.

Bioactivity could be found in either the crude/partition extracts, partially purified extracts or pure form (*Feas et al., 2012*; *Khongkarat et al., 2020*). In the crude partition extracts of this study, only DCMXBP showed anti-amylase activity. With respect to the AChEI activity, only one of the partition extracts showed a weak AChEI activity. For the anti-lipase (as PPLI) activity, all the DCM partition extracts showed PPLI activity, but with only a weak PPLI activity shown by DCMHBP. In contrast, DCMMBP and DCMCBP showed a moderate PPLI activity. Thus, DCM is the best partition solvent for our extraction method. However, these three activities were reported previously in the methanolic extracts of mono- and hetero-floral BP in Brazil (*Araujo et al., 2017*) and the anti-amylase activity was found in the aqueous-ethanolic extract of BP in Nigeria (*Daudu, 2019*). Thus, the solvent and extraction method used could not be ignored.

Among the three bioactivities examined in this study, the PPLI activity of *M. diplotricha* BP became the main interest in this study due to the scarcity of reports on this activity in BP. Thus, DCMMBP from *M. diplotricha* BP was further fractionated by SG60 CC two times. After enrichment, two interesting fractions were derived and the chemical structure of the components in these two fractions was analyzed by [1]H-NMR. One was found to be a mixture of FFAs and showed the highest PPLI activity, while the other fraction was naringenin, a flavanone compound that has been reported in *Mimosa pigra* L. bee pollen (*Khongkarat et al., 2021*). Therefore, it can be used as a marker of *Mimosa* spp. bee pollen.

To study the FFA composition, the fatty acid fraction was esterified to FAME and analyzed by GC-FID following the AOAC method 996.06. The result showed that this fraction consisted of two major PUFAs (linoleic acid at 49.75% and $\alpha$-linolenic at 18.43%)

and one major SFA (palmitic acid at 24.77%). The major fatty acid composition of *M. diplotricha* BP is consistent with the fatty acid compositions that have been reported previously (*Araujo et al., 2017*), except that linoleic acid was found at the highest proportion in our study and not $\alpha$-linolenic acid. It is possible that linoleic acid provided the observed PPLI activity in *M. diplotricha* BP, consistent with the previously reported anti-lipase activity of a fatty acid mixture from *Nigella sativa* extracts with linoleic acid as the dominant fatty acid. This showed a mixed inhibition type (the inhibitor can bind to enzyme whether or not the enzyme has already bound the substrate) (*Shamsiya, Manjunatha & Manonmani, 2016*). The lipase enzyme active site at Ser152 is within a hydrophobic hexapeptide sequence (Val-Gly-His-Ser-Gln-Gly) (*Duan, 2000*), and the long HC chain of linoleic acid can bind with lipase *via* hydrophobic interactions.

Overall, the results indicate that *M. diplotricha* BP has an anti-lipase property due to its FFA composition, which is safe to consume and has the potential to be developed for use as a pharmaceutical supplement.

## CONCLUSIONS

Active biomolecule analysis of BP with PPLI activity contributed to a deeper characterisation of BP. The highest PPLI activity was revealed in *A. mellifera* monofloral MP dominant in *M. diplotricha* floral pollen. Floral identification ensured both the safety for consumption and the standard of quality control. Although naringenin was found not to have a PPLI activity in this work and was also without any antioxidant activity in a previous study, it could be used as a marker for monofloral BP dominant in *Mimosa* spp. The FFAs found in this bee product are here reported to present an *in vitro* lipase inhibitory activity. Thus, this work promotes the use of bee products as a natural nutrient supplement and indicates the benefit of *Mimosa* spp., which are generally regarded as only weeds. However, *in vivo* assessment is still required.

**Abbreviations**

| | |
|---|---|
| **BP1** | bee pollen of *Camellia sinensis* L. |
| **BP2** | bee pollen of *Helianthus annuus* L. |
| **BP3** | bee pollen of *Mimosa diplotricha* |
| **BP4** | bee pollen of *Nelumbo nucifera* |
| **BP5** | bee pollen of *Xyris complanata* |
| **BP6** | bee pollen of *Ageratum conyzoides* |
| **DCMCBP** | DCM partitioned extract of *C. sinensis* L. bee pollen |
| **DCMBP** | DCM partitioned extract of *H. annuus* L. bee pollen |
| **DCMMBP** | DCM partitioned extract of *M. diplotricha* bee pollen |
| **DCMNBP** | DCM partitioned extract of *N. nucifera* bee pollen |
| **DCMXBP** | DCM partitioned extract of *X. complanata* bee pollen |
| **DCMABP** | DCM partitioned extract of *A. conyzoides* bee pollen |
| **HXCBP** | hexane partitioned extract of *C. sinensis* L. bee pollen |
| **HXHBP** | hexane partitioned extract of *H. annuus* L. bee pollen |
| **HXMBP** | hexane partitioned extract of *M. diplotricha* bee pollen |

| | |
|---|---|
| **HXNBP** | hexane partitioned extract of *N. nucifera* bee pollen |
| **HXXBP** | hexane partitioned extract of *X. complanata* bee pollen |
| **HXABP** | hexane partitioned extract of *A. conyzoides* bee pollen |
| **MTCBP** | MeOH partitioned extract of *C. sinensis* L. bee pollen |
| **MTHBP** | MeOH partitioned extract of *H. annuus* L. bee pollen |
| **MTMBP** | MeOH partitioned extract of *M. diplotricha* bee pollen |
| **MTNBP** | MeOH partitioned extract of *N. nucifera* bee pollen |
| **MTXBP** | MeOH partitioned extract of *X. complanata* bee pollen |
| **MTABP** | MeOH partitioned extract of *A. conyzoides* bee pollen |
| **DCMMBP1-5** | pooled fraction 1-5 after SiG60-CC (500-mL size) of DCMMBP |
| **DCMMBP2-1, 2-2, and 2-3** | pooled fraction 1, 2, and 3 after SiG60-CC (250-mL size) of DCMMBP2 |

### Funding

This work was financially supported by the Science Achievement Scholarship of Thailand, the 90th Anniversary of Chulalongkorn University Fund (Ratchadaphiseksomphot Endowment Fund), Toray Science Foundation, and Thailand Science Research and Innovation Fund Chulalongkorn University (CUFRB65_food(6)_114_23_44). The funders had no role in study design, data collection and analysis, decision to publish, or preparation of the manuscript.

### Grant Disclosures

The following grant information was disclosed by the authors:
The Science Achievement Scholarship of Thailand.
The 90th Anniversary of Chulalongkorn University Fund.
Ratchadaphiseksomphot Endowment Fund.
Toray Science Foundation.
Thailand Science Research and Innovation Fund Chulalongkorn University
: CUFRB65_food(6)_114_23_44.

### Competing Interests

The authors declare there are no competing interests.

### Author Contributions

- Phanthiwa Khongkarat performed the experiments, analyzed the data, prepared figures and/or tables, and approved the final draft.
- Prapun Traiyasut performed the experiments, authored or reviewed drafts of the paper, and approved the final draft.
- Preecha Phuwapraisirisan conceived and designed the experiments, analyzed the data, prepared figures and/or tables, authored or reviewed drafts of the paper, and approved the final draft.

- Chanpen Chanchao conceived and designed the experiments, analyzed the data, prepared figures and/or tables, authored or reviewed drafts of the paper, and approved the final draft.

## Data Availability

The raw data for enzyme inhibition and IC50 values and peak data from NMR and chromatogram after purification are available in the Supplemental Files.

## Supplemental Information

Supplemental information for this article can be found online at http://dx.doi.org/10.7717/peerj.12722#supplemental-information.

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
