# Peer review of "First report of fatty acids in Mimosadiplotricha bee pollen with in vitro lipase inhibitory activity"

_PeerJ, doi:10.7717/peerj.12722_

## Round 0.1 · original submission · Major Revisions

Please provide a comprehensively revised version addressing the editorial comments and a detailed rebuttal letter. In particular, professional proofreading service is required since there are English style issues that preclude proper understanding. There is also a reference to some publications that the authors should refer to.

·

Basic reporting

The present study entitled “First report of fatty acids in Mimosa diplotricha bee pollen with in vitro lipase inhibitory activity” by Phanthiwa Khongkarat and colleagues described the chemical composition of different floral bee pollen and the biological activities also been shown. Furthermore, naringenin was identified as a compound marker for Mimosa BP. Despite authors did some interesting work on these pollens, there still have some limitations for this study.
The presentation on the article is not sound and clear. There are too many redundant words in your abstract and introduction, make me sleepy. Only simple and concise writing for an audience lead to a clear understanding on the key information on your study.
Figures and tables must be self-explained, you used too many abbreviations in your tables, figures without necessary explanation. This also makes me puzzling for understanding your findings. I suggest you might take some photos of these plants, and showed them in your background, them give them explanation, and then define the abbreviations. These abbreviations also need to be explained as table note/figure legend.
Figure quality is low. You might chose professional software, like sigmaplot/prism, rather than excel to show the line chart. Some figure can be merged into a big figure, like figure 2,3. They are palynological analysis of pollens. Figure 7, figure 8 just showed chemical structure of known compound, there is no need to present them as separate figure. I suggest authors delete them or list as supporting information.
Authors indicated that “Thus, it can be concluded that naringenin was compound marker for Mimosa BP.” This is incorrect and makes no senses to me. As far as I know, naringenin is a common flavanone which has been widely reported in different types of bee pollen. FYI, I’ve listed some pervious publications, “Nano-liquid chromatography in nutraceutical analysis: Determination of polyphenols in bee pollen; Antioxidant activity of Sonoran Desert bee pollen; Impact of SchisandraChinensis Bee Pollen on Nonalcoholic Fatty Liver Disease and Gut Microbiota in HighFat Diet Induced Obese Mice.”
Other suggestions,
Abstract should be complete rewritten. You might concise it, make it up to 200 words.
Introduction, you need mention the biological activities of bee pollen, and explain the biochemical index you tested means what.
Methods, LN 139- Each BP was stored at room temperature (25 ℃) until used.
Please note that this storage of bee pollen is easily getting spoiled and fatty acids will be oxidized.
The morphology and 152 characteristics of the BPs were recorded and compared to the reported publications.-relative references is missing.
Results,
LN372 In order to confirm the plant origin of the BP, all samples were additionally identified by molecular analysis-suggest delete.
LN286 Thus, the six BPs were confirmed to be essentially monofloral, matching the principal flowers around the hives, and with the morphological and molecular analyses congruent with each other. -Such words can be moved to discussion part.
LN400- In this study, the inhibitory activity against.. DELETE
LN489- This is the first report that fatty acids in BP have a PPLI activity. In addition, naringenin can be used as a chemical marker for Mimosa bee pollen. DELETE. Also, such words should be toned town.

Experimental design

See above

Validity of the findings

See above

Additional comments

See above

Reviewer 2 ·

Basic reporting

The English language used was intelligible, but the paper requires a major reconstruction in a way that the facts are logically presented. The background/context is not really stated properly and the research aim is not backed up by research gaps. Spelling errors were also observed as well as some grammatical errors. The author can easily address this by reconstructuring the introduction part and then clearly indicate the gaps of research.

Experimental design

The research question was not defined properly. a lot of information are lacking particularly the reagents and chemicals section, the model and brands for the equipment used. The authors had only performed one replicate extraction for all analysis. The authors has to include the said information and the justification on why they only performed one replicate for the extraction.

Validity of the findings

The authors had only performed one replicate extraction for all the assay analysis. the authors had also not indicated if whether the assays used were validated based on the ICH guidelines.
The result of the TLC analysis for naringenin has to be improved. include the Rf values of the standard and the unknown and if possible run it in a solvent of higher polarity so that the compounds will be sitting in the middle of the plate. the NMR analysis of the compounds are insufficient to prove that the compounds analysed were actually fatty acids. the fatty acid determination using the GC-FID is promising but i suggest that the authors should also add more verification for this. Quantification can also be done for the fatty acids so that the claims of detecting it in the be pollen samples can be more valid.

Additional comments

I commend the authors for using a bioassay guided isolation technique which is very helpful and useful to ensure that what ever secondary metabolite obtained is bioactive and of significance.
The abstract is very informative and had covered all the necessary information for a summary, however, it is too long. For the abstract, you do not need to introduce if what bee pollen is and can leave it to the introduction. You can start with the aim of the study instead. Also, you may not need to mention the other species of bee pollen floral source and focus only on Mimosa diplotricia.
The introduction is informative and have covered most of the information required but it has to be improved by presenting the facts more logically. Each paragraph must focus only on a certain topic until it builds up into the research gap and research aim. Furthermore, the introduction should add more information to justify if why the research needs to be performed (for example, the extent of research done on the bee pollen used in the study both in Thailand and abroad). You may also remove the details about what was done in the study and just focus on the discussion of the research gaps that prompted you to perform the research.
For the materials and methods, you need to include a section that indicates the chemicals and reagents (supplier, city and country) that were used in the study. Also, you needed to specify the name of all the equipment, model, city, and country where it was manufactured.
I have a question about the bee pollen, was it dried prior to storage at room temperature? And for ow long was it stored prior to use?
Have you performed an optimisation study on the solvent parameters used to extract the bee pollen? If not, you need to cite a reference for the ectraction method that you used.
How many replicates of bee pollen were extracted?
What were the concentrations used for the assay? this must be included in the methodology and not in the results.
Clarify the solvent used for the TLC analysis of naringenin. How much is the concentration of DCm and Methanol?

·

Basic reporting

The manuscript "First report of fatty acids in Mimosa diplotricha bee pollen with in vitro lipase inhibitory activity" is written in a clear and orderly way. Supported by references on the subject. The information presented in the tables is clear and sufficient to show the results and the figures are of the necessary quality for publication.

Experimental design

The materials and methods presented are clear and adequate and presented in detail to be reproducuble by anothers.

Validity of the findings

The results are validate with the information generated and showed in the supplementary material.

The conclusions are connected to research and are limited and supported by results and experiments.

Additional comments

Line 90- change to "activities in"
Line 341 It is suggested to use statistical software for the determination of means and standard deviations.
Line 426. What does "reasonable" mean. Explain.

---

## Round 0.2 · Minor Revisions

Please provide a revised version addressing the editorial comments and a rebuttal letter.

Reviewer 2 ·

Basic reporting

English editing has to be done in order to have a more understandable and coherent paper.

Experimental design

The researchers has to indicate that the pollen samples used were dried using a specific process.

Validity of the findings

see above

·

Basic reporting

Comments were answered

Experimental design

Comments were answered

Validity of the findings

Comments were answered
Only
Line 426. What does "reasonable" mean. Explain.

Authors We changed it to be “satisfying” instead. It means that this concentration was acceptable to use for the primary screening.

This "reasonable" or "satisfying" concentration I suggest expressing in some kind of unit.

Additional comments

The authors of the article: "First report of fatty acids in Mimosa diplotricha bee pollen with in vitro lipase inhibitory activity", comprehensively answered the comments made by the reviewers.

---

## Round 0.3 · accepted · Accept

Thanks for addressing all the revisions and corrections requested. Now your manuscript is accepted in PeerJ.